# Effects of Platform Pre-Heating and Thermal-Treatment Strategies on Properties of AlSi10Mg Alloy Processed by Selective Laser Melting

**Riccardo Casati** [1] , **Milad Hamidi Nasab** [1] , **Mauro Coduri** [2] , **Valeria Tirelli** [3] **and Maurizio Vedani** [1,*]

1   Department of Mechanical Engineering, Politecnico di Milano, 20156 Milan, Italy; riccardo.casati@polimi.it (R.C.); milad.hamidi@polimi.it (M.H.N.)
2   ESRF, European Syncrotron Radiation Facility, 38043 Grenoble, France; mauro.coduri@esrf.fr
3   Aidro S.r.l, 21020 Taino, Italy; valeria.tirelli@aidro.it
*   Correspondence: maurizio.vedani@polimi.it; Tel.: +39-02-2399-8230

**Abstract:** The AlSi10Mg alloy was processed by selective laser melting using both hot- and cold-build platforms. The investigation was aimed at defining suitable platform pre-heating and post-process thermal treatment strategies, taking into consideration the peculiar microstructures generated. Microstructural analyses, differential scanning calorimetry, and high-resolution diffraction from synchrotron radiation, showed that in the cold platform as-built condition, the amount of supersaturated Si was higher than in hot platform samples. The best hardness and tensile performance were achieved upon direct aging from cold platform-printed alloys. The hot platform strategy led to a loss in the aging response, since the long processing times spent at high temperature induced a substantial overaging effect, already in the as-built samples. Finally, the standard T6 temper consisting of post-process solution annealing followed by artificial aging, resulted in higher ductility but lower mechanical strength.

**Keywords:** selective laser melting; AlSi10Mg alloy; processing temperature; aging treatment

## 1. Introduction

Due to large freedom in shaping and high material usage efficiency, selective laser melting (SLM) is attracting a great interest for a large number of industrial applications requiring near-net shape manufacturing of light components. SLM allows generating parts by selectively scanning powdered metals with a laser beam, so as to produce layer by layer solid volumes after melting and solidification [1,2]. AlSi10Mg and AlSi7Mg alloys are amongst the most popular materials for light structures built by SLM. They are selected among the traditional foundry alloys featuring good castability and well-known solidification behavior [3–5].

The trend toward increased lightness and higher load-bearing capacity is now pushing for higher material strength, which can be achieved either by alternative Al alloy formulations or by optimized thermal treatments of current materials. High-strength precipitation hardenable Al-Zn-Mg-Cu and Al-Cu-Mg-Si alloys offer large opportunities to achieve high mechanical performances, but their processability is often limited due to hot-cracking phenomena occurring during the last stages of the rapid solidification induced by SLM [6,7]. An alternative approach, aimed at raising the strength of the above easily processable Al-Si-Mg alloys, is to optimize their post-process thermal treatments, trying to exploit, as much as possible, the supersaturation of the solid solution and the fine cellular structure brought about by the rapid solidification and cooling induced by SLM.

From the recent literature, there is evidence that several kinds of post-process heat treatments can be considered. Stress relieving is often carried out on as-built AlSi10Mg parts at temperatures of 300 °C for 2 h [8–11], but investigations showing the effectiveness of annealing treatments at temperatures as low as 200 °C are also available [10,12]. Such treatments lead to a substantial increase in alloy ductility and a concurrent drop in strength.

Solution-annealing treatments and solution annealing followed by aging (T6-like thermal treatments) have been also investigated on AlSi10Mg and AlSi7Mg alloys processed by SLM, in order to modify the properties of the as-built materials [3,5,9,11,13–19]. It was shown that the standard solution annealing, followed by water quenching and artificial aging, replaces the fine α-Al cells, and the Si network segregated at their boundaries into coarser grains and globular Si particles. Evidence of laser tracks, and of the related heat-affected zones, also disappeared during solution annealing [3,9,13,15,19]. Also, in this case, fracture elongation significantly improves, but hardness, yield strength, and ultimate tensile strength drop down to values lower than those of the as-built material.

An interesting database about the mechanical properties available in literature for the AlSi10Mg alloy, processed by SLM, was given in a recent paper by Tang and Pistorious [20], who showed a large variability of the achieved performance depending on sample orientation, processing parameters, and heat treatment condition. Maamoun and co-workers [21] published a research study on thermal post-processing of AlSi10Mg alloy after SLM, and supplied conclusive suggestions in the form of a microstructure/microhardness map on possible options for post-processing treatments. However, it must be considered that the adoption of the aging treatment was evaluated only after the standard solution-annealing treatment, according to conventional T6-like treatment used in cast and wrought Al alloys.

The aging response of an A357 (AlSi7Mg) alloy produced by SLM was specifically investigated in [18]. Differential scanning calorimetry (DSC) and microhardness analyses showed that SLM-processed samples feature the same precipitation sequence of the corresponding cast alloy, by the sequential formation of Mg-Mg, Si-Si, and Mg-Si co-clusters, β″, β′, and β ($Mg_2Si$) phases, along with the precipitation of Si [22–24]. Moreover, it was demonstrated that the as-built and the solution-annealed and quenched samples experienced, substantially, the same precipitation sequence when subjected to direct aging. The as-built condition can, therefore, be considered as fully appropriate for an effective dispersion strengthening process by direct artificial aging, without the need of any post-SLM solution-annealing treatment.

Further information on aging behavior of supersaturated alloys generated by rapid solidification can be obtained from a recent work published by Marola and co-authors [25]. Their comparative study on an AlSi10Mg alloy, processed by SLM, copper mold casting, and melt spinning, showed that extensive Si supersaturation takes place in rapidly solidified alloys. At the early stages of aging, the supersaturated Si was able to precipitate, leading to broad DSC signals that overlap to those originally assigned to β″ formation. These results are consistent with XRD data supplied by Li and co-workers in their study on effects of heat treatments of AlSi10Mg alloy produced by SLM [13].

The above observations suggest that the peculiar microstructure of Al alloys after the rapid solidification and cooling conditions inherited by SLM could be considered as a profitable starting point for the design of optimized aging treatments in additively manufactured Al alloys. The present paper is, therefore, aimed at evaluating the benefits of different possible heat treatment strategies on an AlSi10Mg alloy. Opportunities offered by SLM processing on room-temperature building platforms, rather than on hot building platforms, are also investigated. In the first scenario, the supersaturated solid solution of as-built material is exploited as the starting condition for subsequent aging whereas, in the latter, the strengthening action of a possible in situ aging treatment is investigated.

## 2. Materials and Methods

An EOS M290 SLM system was adopted to process gas-atomized AlSi10Mg powders according to the parameters supplied in Table 1. The table also shows that a set of samples, labeled as HP,

was produced by depositing powder layers on a preheated build platform at 160 °C, whereas a second set, labeled as CP, was printed without any preheating. The specific temperature level of 160 °C, for the hot platform, was selected in order to reproduce the usual aging temperature adopted for cast AlSi10Mg alloys. It should be considered that information about the build platform temperature are not always supplied in the experimental descriptions of published investigations. Examples of 35 °C [8,10,26–28], 100 °C [13], and 200 °C [21] platform processing are available in literature.

Cubes of 20 mm side, and cylindrical samples with a diameter of 10 mm and length of 100 mm, were built using a scanning strategy involving a rotation of 67° between consecutive layers. Sample density was checked by Archimedes' method, assuming a theoretical density of the alloy of 2.67 g/cm$^3$. Five distinct measurements were performed.

**Table 1.** Process parameters adopted for the production of the AlSi10Mg samples.

| Sample Code | Power (W) | Hatch Distance (mm) | Scan Rate (mm/s) | Layer Thickness (mm) | Platform Temperature (°C) |
|---|---|---|---|---|---|
| CP | 340 | 0.2 | 1300 | 0.03 | RT |
| HP | | | | | 160 |

Different thermal treatment conditions were considered for the investigation, according to data collected in Table 2. In particular, samples produced by the cold platform were investigated either in the as-built (AB), as-built and directly aged (T5 temper), and solution-annealed, water-quenched, and aged (T6 temper) conditions. Samples produced by the hot platform procedure were only considered in the as-built condition, in an attempt to investigate the possible in situ aging effects induced by the high-temperature holding times during SLM processing.

Scanning electron microscope (SEM) (EVO 50, Zeiss, Germany) and optical microscope (Leitz Aristomet, Germany) were used to investigate alloy microstructure after chemical etching with Keller's and Weck's solutions. DSC analyses were performed in an Ar atmosphere using a heating rate of 30 °C/min to investigate aging behavior of the alloy treated according to the different conditions (i.e., as-built from cold platform, as-built from hot platform, after solution annealing and water quenching).

**Table 2.** Investigated tempers of the AlSi10Mg samples (w.q.: water quenching).

| Sample Code | Platform Temperature | Solution Treatment | Aging |
|---|---|---|---|
| CP AB | RT | None | None |
| CP T5 | RT | None | 160 °C 4 h |
| CP sol | RT | 540 °C 1 h, w.q. | None |
| CP T6 | RT | 540 °C 1 h, w.q. | 160 °C 4 h |
| HP AB | 160 °C | None | None |

XRD investigations were carried out at the ID22 beamline of the European Synchrotron Radiation Facility (ESRF) in Grenoble (France), using a high-resolution setup, i.e., with nine scintillators preceded by Si 111 crystal analyzers, which guarantee accurate determination of lattice parameters, as well as suppression of parasitic scattering. The specimens were cut with a cylindrical geometry and diameter of about 1 mm, and rotated during acquisition to increase statistical averaging of crystallites. Incident wavelength was set to 0.35434 Å (~35 keV). XRD patterns were collected, summing different scans with 45 min total acquisition time for each sample, from 0° to 35° in 2 theta. For data analysis, the amounts of Si phase and the estimated standard deviations were computed by Rietveld refinements based on the relative intensities of Si and Al Bragg reflections. Moreover, the crystallite size of Si (when in nanocrystalline form) was evaluated on the basis of the broadening of Bragg peaks, using the Williamson-Hall approach [29].

Tensile tests were performed according to EN ISO 6892-1:2016 standard at room temperature, with a crosshead speed of 0.5 mm/min, using an MTS Alliance RT/100 universal testing frame

equipped with an extensometer. Specimens having a gauge length of 30 mm and a diameter of 6 mm, machined from bars built along horizontal (powder bed layers parallel to longitudinal axis of the specimens) and vertical positions, were adopted. At least three specimens for each condition were tested.

## 3. Results

### 3.1. Microstructure

Representative images of the as-built microstructure of samples, produced by cold and hot platform, are depicted in Figures 1 and 2, respectively. As expected, in both conditions, the distinct tracks of the melt pools left by the laser scanning are visible, especially from lateral views (Figures 1 and 2). No significant differences are detectable in terms of size and shape of the melt pools, when comparing hot and cold platform processing.

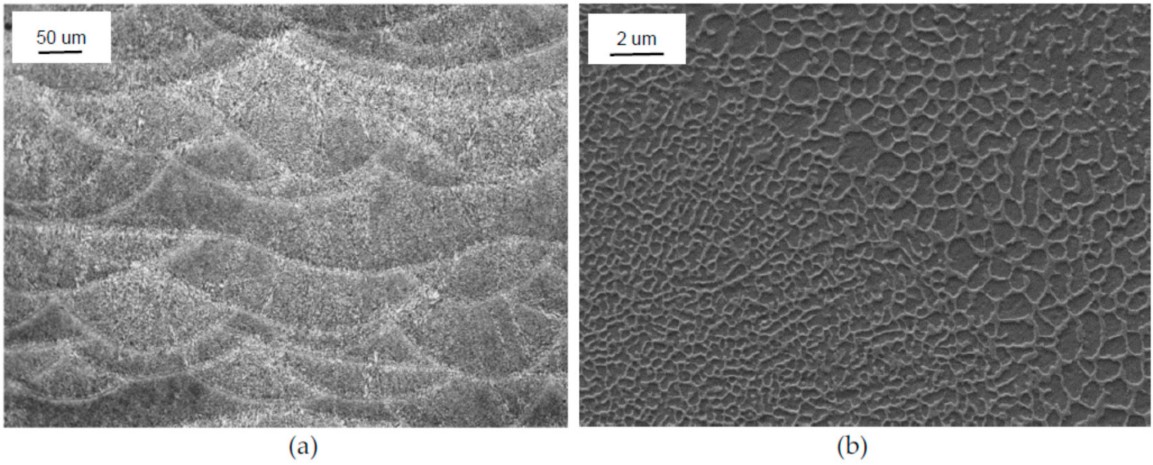

**Figure 1.** Microstructure of the AlSi10Mg alloy produced by selective laser melting (SLM) on a cold platform. (**a**) Low magnification optical microscope view (section plan parallel to building direction) of the solidified melt pools left by laser tracks; (**b**) SEM image of the cellular structure.

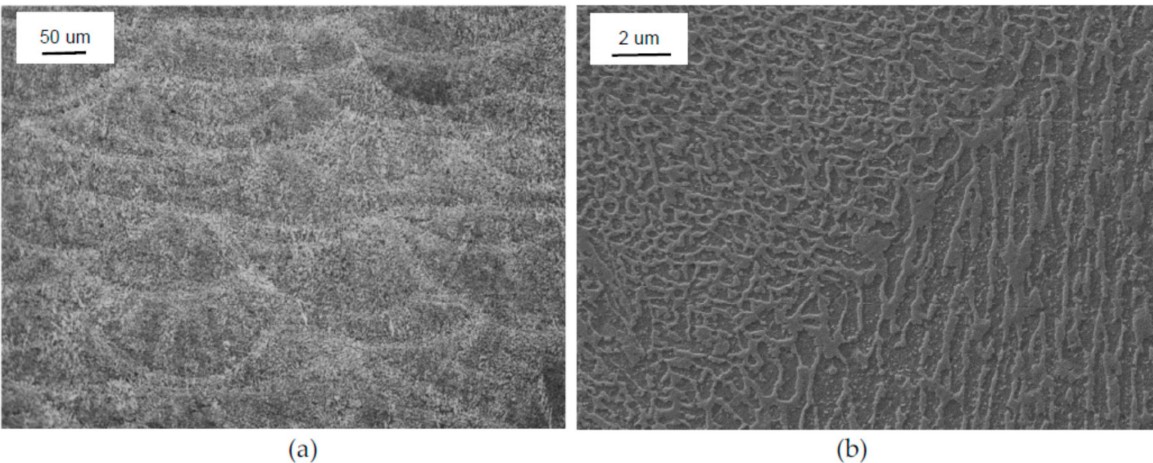

**Figure 2.** Microstructure of the AlSi10Mg alloy produced by SLM on a hot platform. (**a**) Low magnification optical microscope view (section plan parallel to building direction) of the solidified melt pools left by laser tracks; (**b**) SEM image of the cellular structure.

In both cases, a very limited number of defects (oxides, gas pores, unmelted particles, lack of fusion at melt pool boundaries) was detected. The material density, as measured by Archimedes' method, supplied an average value of 99.467% and a standard deviation of 0.004.

High magnification observations revealed, in both conditions, the presence of a very fine cellular solidification structure consisting in submicrometric primary α-Al cells decorated by a network of Si. Figures 1 and 2 also show that, when crossing the boundary of the melt pools, the size of the cells is subjected to a clear change, due to reheating effects produced by the overlaying passes [4], and to local changes in solidification conditions [13]. The comparison between cold and hot platform microstructure, proposed in Figure 3, highlights, by higher magnification views, that the Si network is substantially continuous in the CP samples (Figure 3a), while it becomes broken, combined with an additional intracellular precipitation of tiny particles, in HP samples (Figure 3b). It is presumed that the longer holding period at high temperature results in higher diffusion, leading to spheroidization phenomena in the Si network and precipitation of supersaturated Si atoms at cell interiors. This hypothesis is in good agreement with the results published by Li and co-authors, who investigated the precipitation of Si atoms into nano- or micrometric particles from supersaturated solid solutions in a SLM-processed AlSi10Mg alloy [13].

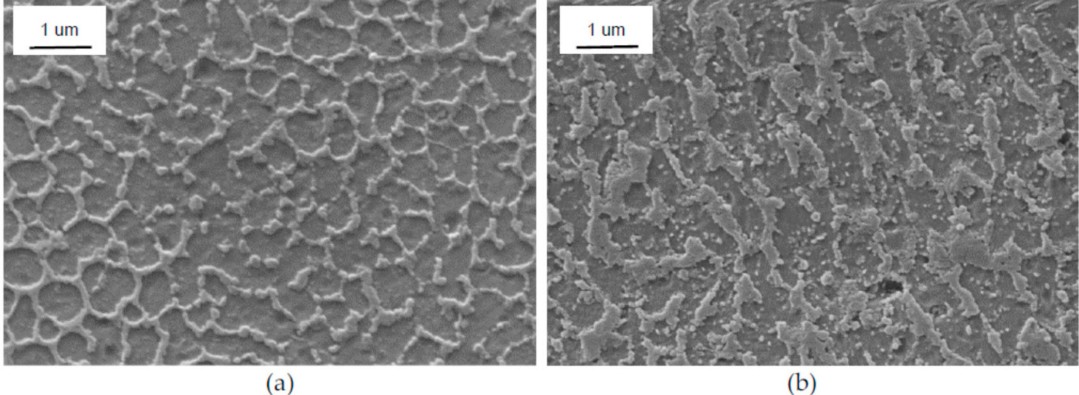

**Figure 3.** High magnification SEM views of the as-built microstructure detected in cold platform (**a**) and hot platform (**b**) samples. The Si network surrounding the α-Al cells is the lighter phase, discrete intracellular Si precipitates are observed in panel b as spheroidal-shaped particles located inside the α-Al cells.

The effects of thermally activated diffusion become more evident after the solution-annealing treatment in CP sol and CP T6 samples. Figure 4 depicts the coarse grain structure formed after holding the alloy at 540 °C for 1 h. It is revealed that the cellular structure fully transforms into a set of equiaxed grains having a size of the order of 10 μm. The Si particles also coarsen, reaching an average size of 2.8 μm and a more rounded shape.

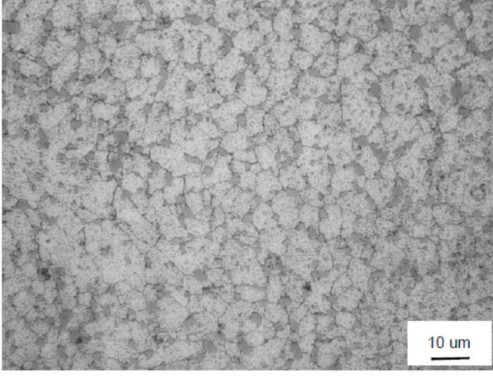

**Figure 4.** Representative microstructure of the solution-annealed (CP T6) sample.

## 3.2. X-ray Diffraction

High-resolution diffraction from synchrotron radiation was mainly used to investigate the evolution of the Si phase, based on phase fraction and lattice parameters. Only Al and Si phases were observed within the detection limit of the instrument. Figure 5 depicts the low-angle region of the patterns of the investigated samples treated according to the different tempers, while Table 3 summarizes the lattice parameters of the $\alpha$-Al and Si phases.

Both the solution-annealing treatment (sample CP sol and CP T6) and the hot platform solidification mode (sample HP AB) promote higher fraction of precipitated Si. In CP AB sample, a significantly lower amount of Si is detected, but it readily increases after aging (sample CP T5). Consistently, the lattice parameters of the $\alpha$-Al show larger distortions with increasing degree of Si supersaturation. It is worth remarking that the reference lattice parameters of the $\alpha$-Al phase in the AlSi10Mg alloy under equilibrium is 0.40515 nm and, by increasing the fraction of solute Si, it is expected to decrease according to the smaller atomic radius of Si, compared to that of Al [25]. As the high resolution of the instrument allows a very accurate determination of lattice parameter of microcrystalline Al, its evolution confirms the trend observed for the Si phase fractions reported in Table 3.

The XRD data also show that the size of precipitates increases when moving from CP AB to CP T5 and HP AB samples, according to a presumable dependence on heating time. Considering the two material conditions involving high-temperature solution treatment (CP sol and CP T6), the breadth of the Si peaks becomes similar to that of $\alpha$-Al, confirming that crystals coarsened to a size within the micrometer scale, too big to be accurately quantified through powder diffraction. The size of 2–8 $\mu$m, already evaluated by microscopy, should be considered for these conditions.

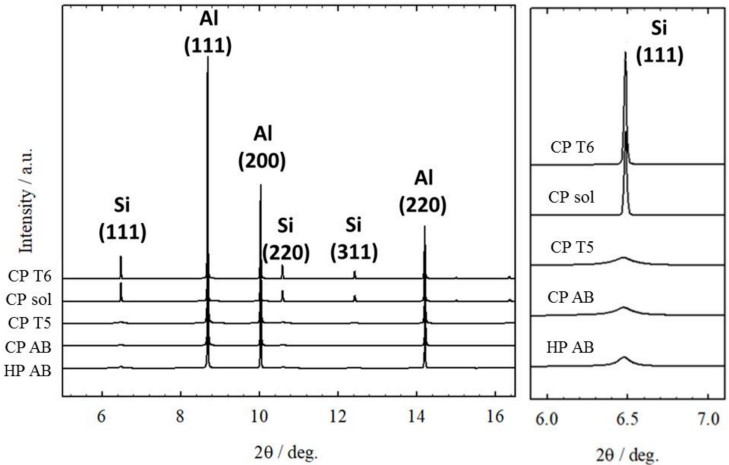

**Figure 5.** XRD patterns of the AlSi10Mg samples treated according to the different tempers. The panel on the right better highlights the changes in peak shape of Si (111).

**Table 3.** Al and Si lattice parameters and features of Si phase detected by XRD (the standard deviation of measures is given in brackets).

| Sample Code | Lattice Parameters (nm) | | Si Phase (wt %) | Size Range of Si Crystallites |
|---|---|---|---|---|
| | $\alpha$-Al | Si | | |
| CP AB | 0.404833 | 0.54361 | 8.77 (0.05) | 8 nm |
| CP T5 | 0.405011 | 0.54386 | 10.03 (0.05) | 8 nm |
| CP sol | 0.405182 | 0.54276 | 10.16 (0.03) | micrometric |
| CP T6 | 0.405155 | 0.54286 | 10.23 (0.03) | micrometric |
| HP AB | 0.405079 | 0.54333 | 10.21 (0.05) | 15 nm |

### 3.3. Thermal Analysis and Aging Curves

DSC curves of as-built and of solution-annealed samples have been collected in order to evaluate the aging potential of the AlSi10Mg alloy, starting from different conditions. Figure 6 displays representative profiles of the heating ramps within the temperature range, where precipitation of strengthening phases is expected. The solution-annealed sample (CP sol) shows the sequence of precipitation peaks according to accepted literature [18,22–25]. The DSC curve of the as-built sample produced on a cold platform (CP AB) shows a similar shape, suggesting that most of the strengthening potential of the alloy can be exploited by thermal aging, starting from the as-built state, without the need of any prior solution-annealing treatment. On the contrary, the curve of the as-built sample produced on hot platform (HP AB) only shows a very limited response to the thermal ramp, confirming that most of the precipitation already occurred during the hot-stage manufacturing process.

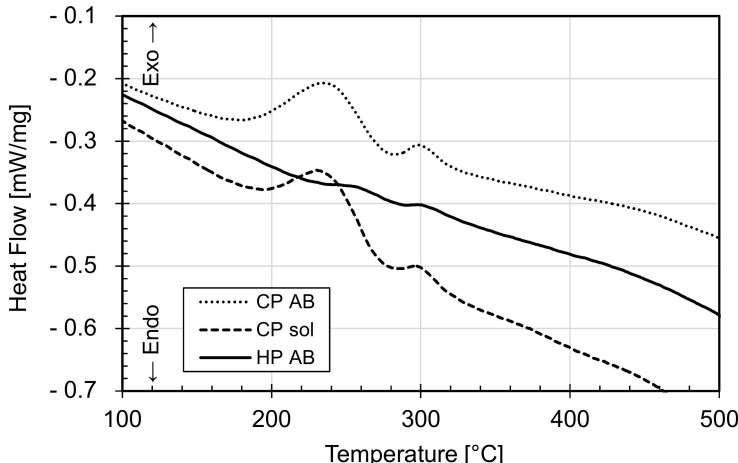

**Figure 6.** DSC scan profiles collected on heating of the SLM-processed AlSi10Mg alloy within the temperature range of interest for the precipitation of the strengthening phases.

The evolution of Vickers' microhardness of as-built samples, as a function of holding time at the aging temperature of 160 °C, is depicted in Figure 7. In full agreement with DSC data, it is reported that aging starting from CP AB condition produces a clear hardening, with an increase in microhardness from 116 to 134 HVn. Conversely, aging of the hot platform-processed samples led to a slight loss in hardness, from 117 to 113 HVn, after 4 h of aging at 160 °C. Thus, it can be reasonably inferred that further aging of HP AB alloy can only induce stress relaxation and/or overaging effects, without any improvement in mechanical properties.

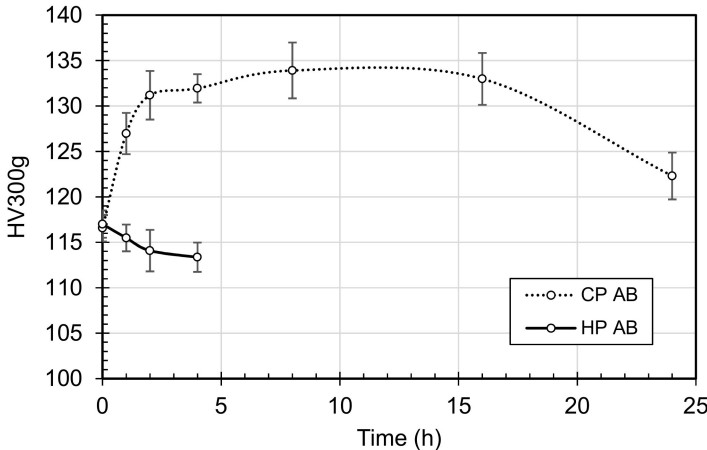

**Figure 7.** Aging curves at 160 °C of as-built SLM-processed AlSi10Mg alloy.

*3.4. Mechanical Properties*

Tensile tests were performed on the investigated alloy to evaluate the influence of the different tempers. In addition, the effect of sample orientation was considered by testing samples machined from both horizontal and vertical bars, namely, specimens with longitudinal axis normal and parallel to the building direction, respectively. Table 4 summarizes the main tensile properties achieved, while Figure 8 shows the recorded tensile curves.

The collected data clearly show that the best strength is achieved by optimally aging the cold platform as-built samples (CP T5), thus preserving the ultrafine cellular structure and inducing the expected precipitation of strengthening phases. In spite of its large aging potential, highlighted by the DSC analyses, the solution-treated samples (CP T6) attained the lowest strength, but took benefit from a remarkable improvement in fracture elongation. Finally, the performance of samples built on hot platform (HP AB) showed an intermediate condition in terms of strength and ductility. It is suggested that, in this temper, the material could benefit from the fine cellular structure, but could not fully exploit its strengthening potential, due to the lack of a tailored aging stage.

Finally, as far as the effect of specimen orientation is concerned, there is evidence from tensile data that CP T5 and HP AB samples tested in the horizontal direction always feature higher yield strength and fracture elongation, but lower ultimate tensile strength. On the contrary, solution annealing and aging (CP T6) led to the best yield and ultimate strength values in the horizontal samples, but comparable ductility for the two directions.

**Table 4.** Average tensile data (UTS: ultimate tensile strength; YS: yield strength) of the SLM-processed AlSi10Mg alloy heat-treated to the different tempers. Standard deviation is given in brackets.

| Sample Code | Sample Orientation | UTS (N/mm$^2$) | 0.2 YS (N/mm$^2$) | Strain at Fracture (%) |
|---|---|---|---|---|
| CP T5 | horizontal | 471 (0.8) | 321 (1.8) | 8.6 (0.5) |
| | vertical | 493 (0.6) | 292 (0.6) | 6.0 (0.6) |
| CP T6 | horizontal | 323 (0.0) | 243 (0.0) | 15.3 (2.4) |
| | vertical | 302 (1.4) | 223 (2.8) | 16.0 (2.5) |
| HP AB | horizontal | 386 (2.6) | 248 (1.7) | 8.6 (1.4) |
| | vertical | 412 (5.5) | 228 (4.1) | 7.0 (0.1) |

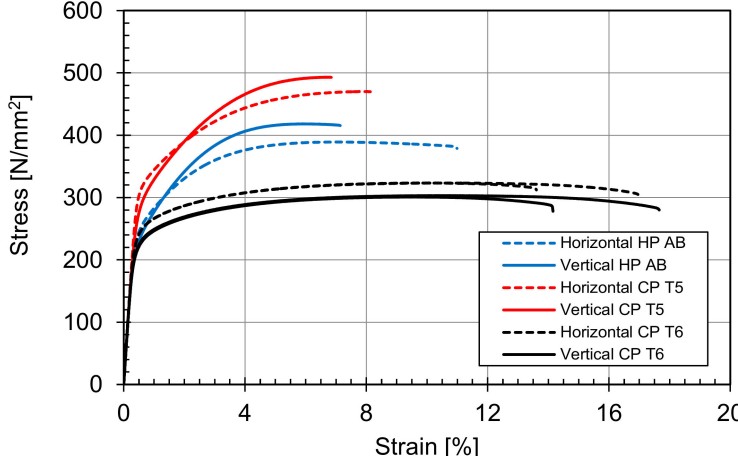

**Figure 8.** Tensile curves of the SLM-processed AlSi10Mg alloy heat-treated to the different tempers.

## 4. Discussion

The present results clearly demonstrate that optimization of thermal treatment of SLM-processed Al alloy can led to a remarkable improvement in strength over the as-built condition, and that the standard thermal treatment procedures (i.e., T6 temper) might be inadequate when trying to improve

the tensile strength. Skipping the solution-annealing stage and promoting precipitation aging, starting right from the supersaturated, rapidly solidified and cooled, as-built material, already revealed itself to be a promising strategy for other alloys [30–32]. Jägle and co-workers [30] deeply investigated the effects of the thermal cycles experienced in various age-hardenable alloys during SLM and laser metal deposition. Their study highlighted desired and undesired precipitation reactions that could occur on the processing of several alloys, during intrinsic heat treatments generated by subsequent deposition passes (i.e., in already consolidated material) and during the following aging treatments. It was concluded that laser additive manufacturing can be used to produce supersaturated alloys that can lead to precipitation of second phases during a following aging stage. The present findings confirm that, also in the AlSi10Mg alloy, the precipitation reaction can be effectively exploited starting from the as-built condition, and it can lead to an appreciable increase in mechanical strength, reaching values of 471 MPa and 321 MPa for the ultimate tensile strength and the yield strength of the CP T5 samples, respectively. When compared to the tensile data survey given in the literature [20], the achieved performance ranks in the highest positions, also considering the favorable trade-off with fracture elongation. Conversely, the inappropriateness of a conventional T6 heat treatment on SLM-processed alloys is highlighted, owing to extensive grain growth induced by the high-temperature solution annealing and coarsening of the Si particles. A careful observation of the tensile curves reveals that the samples tested in their original SLM solidification structure (i.e., CP T5 and HP AB samples) show distinctive shapes of the stress vs strain curves. Specimens machined from horizontal bars have higher yield strength, and lower work-hardening rate and ultimate tensile strength, combined with improved ductility (see Table 4). Such behavior is supposed to be due to peculiar crystal orientation of the elongated solidification grains (containing the above described cellular substructure) that produce an anisotropic effect in material behavior [4,19], and to possible preferential location of small defects at laser track boundaries, leading to easier crack growth normal to loading direction, in vertical specimens.

　　A further issue to be considered when comparing cold and hot platform-printing strategies is the effectiveness of the in situ aging process in the latter situation. From DSC and microhardness results, it is inferred that HP AB samples do not show any significant response to aging under DSC heating ramps, and are even subjected to a slight loss in hardness on isothermal aging (see Figure 7). This suggests that overaging effects could take place in the samples, and it is reasonable to speculate that the actual aging effect promoted by hot platform processing would depend on printing time and on the position of the reference volume along the part height. Indeed, material volumes close to the build platform are subjected to longer processing times than those located close to the upper surfaces of the part. The volume of printed parts should also play a role, since thin parts featuring a lower re-melted area and higher fraction of relatively cold powder surrounding it would experience shorter holding times at high temperature than bulky shapes. A simple experimental test was designed in this perspective, measuring the microhardness profile along a sectioned vertical cylindrical bar (10 mm in diameter, 100 mm in length) produced by the hot platform strategy. The profile, reported in Figure 9, shows that no significant trend can be envisaged, at least for the bar size and the alloy investigated here. A similar experiment was also performed by Maamoun and co-authors [21], who showed, for the same AlSI10Mg alloy, an increasing hardness trend with increasing distance from platform. However, their samples were only 10 mm high along the z direction, and the SLM process was performed by a build platform temperature of 200 °C, instead of the 160 °C used in the present investigation.

　　It must also be considered that hot platform processing is of paramount importance for the control of residual stresses and distortions for difficult-to-process alloys and complex geometries [10,33–35]. Therefore, the hot platform processing method could represent a profitable alternative in the case of large parts with critical shapes, also considering cost savings related to the absence of any thermal post-processing stage.

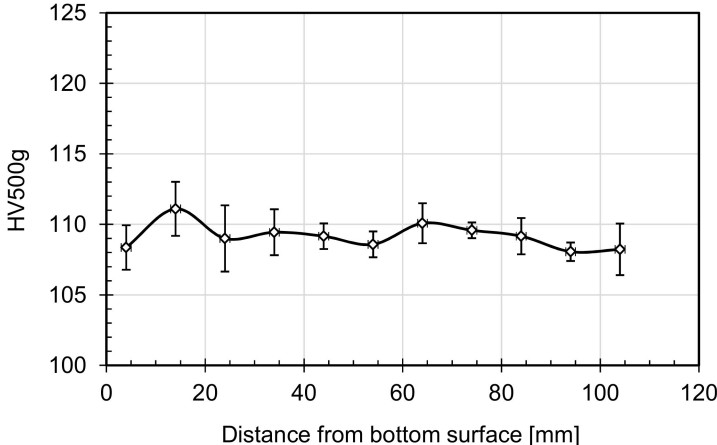

**Figure 9.** Microhardness profile measured in a HP AB sample as a function of position along build direction. Data refers to average and standard deviation values out of five microhardness readings per position.

## 5. Conclusions

Hot and cold platform processing were considered, among other strategies, to assess the aging response of the AlSi10Mg alloy processed by SLM, in order to achieve improved mechanical properties. In particular, hot platform was devised to test the ability of the alloy to age harden "in situ", namely, during the printing process itself. On the contrary, cold platform was used in order to investigate the possibility of directly aging the alloy right after the SLM process, without the need of a pre-solution-annealing treatment. The achieved results allowed for drawing the following conclusions.

The microstructure of the alloy produced by SLM on a cold platform consists of ultrafine $\alpha$-Al cells surrounded by a network of Si phase. High-resolution XRD analysis demonstrated that, in the cold platform as-built condition, the amount of supersaturated Si is higher than in hot platform and in solution-annealed samples.

Upon aging, the rapidly solidified cold platform-processed alloy could be directly strengthened starting from as-built condition, supplying the highest level of hardness and tensile strength, together with fairly good levels of fracture elongation.

Hot platform processing led to a significant loss of the aging response with a concurrent drop of mechanical properties. It is supposed that the rather long holding times spent at high temperature during processing could induce a substantial overaged temper in the HP samples.

The standard solution annealing, followed by water quenching and artificial aging, resulted in the lowest achievable strength, thus demonstrating the inappropriateness of a conventional T6 heat treatment on SLM-processed AlSi10Mg alloys when high-strength alloys are required.

**Author Contributions:** Conceptualization by R.C., M.H.N., M.V.; V.T. supplied the SLM samples, M.C. conducted XRD experiments at ESFR, and performed data analysis; M.H.N. performed microstructural analyses, M.V. wrote the manuscript. All authors discussed the results and approved the final manuscript.

**Funding:** The present research work has been partially funded by Regione Lombardia within the frame of the project "NUVOLE" (id 187033) for Politecnico di Milano and within the project INNODRIVER-S3—2017 for Aidro S.r.l.

**Acknowledgments:** Research activities have been supported by the use of the interdepartmental Laboratory AMALA at Politecnico di Milano. The European Synchrotron Radiation Facility in Grenoble (France) is acknowledged for provision of beamtime.

**Conflicts of Interest:** The authors declare no conflict of interest.

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
