# Peer review of "Effects of Platform Pre-Heating and Thermal-Treatment Strategies on Properties of AlSi10Mg Alloy Processed by Selective Laser Melting"

_metals, doi:10.3390/met8110954_

Reviewer 1 Report

The manuscript presents an interesting study of the effect of the thermal exposure on the microstructure and mechanical properties of AlSi10Mg processed by selective laser melting SLM.

The paper reported an original work that merit to be published.

However, the manuscript presents several points that need to be discussed before considering it for publication.

1)     The authors need to expand the literature on heat-treated AlSi10Mg produced by SLM.

In order to provide a better background to this topic, several papers can be used

2)     The authors should report the T5 and T6 conditions used.

3)     The authors should explain clearly the reason why pre-heating at 160 °C was used.

The ageing treatment of T6 is carried out at 160 °C, but the authors should clearly explain this point in order to improve the understanding of the temperature selection.

4)     The data should be compared with the standard pre-heating temperature recommended by EOS. The authors may use the available literature to reveal similarities and difference with the current results.

5)     The part on the microstructure analysis required to be explained more in details (3.1 Microstructure).

 ·      Please report the level of densification level of the samples.

 ·      Please provide SEM images at lower magnification in order to better show the difference between AlSi10Mg built on cold platform and AlSi10Mg built on hot platform (Figure 1 and Figure 2).

  ·     The authors may provide some measure for the cellular structures comparing it with the literature.

6)     I would recommend to compare the mechanical properties with as-built and heat-treated AlSi10Mg alloy produced by SLM reported in the literature, providing a more extensive comparison.

Author Response

The authors would like to thank for the useful suggestions received that, in our hope, significantly contributed to improve the quality of the manuscript and to fill some gaps that were actually present in its first version.
Since several comments were raised by more than one reviewer, we prepared a single response letter were all the answers are collected.

Reviewer 2 Report

The present paper "The effect of print and thermal-treatment strategies on properties of SLM-processed AlSi10Mg alloy" contains relevant and interesting results in the field of additively manufactured Al-based materials. However I have following remarks for this publication:

1.)  Lines 102-106: 

    I think it is important to add the information of the standard (DIN/ ASTM) that was used for conduction of the tensile tests.

 2.) Lines 127-128:

    I think the authors want to refer to Figure 1b and Figure 2b instead of Figure 1a and Figure 2a.

 3.) Table 3:

    For the interpretation of the results of tensile tests it would be very helpful to add a statistical deviation of the content of the Si-Phase. Furthermore a quantification of the sizes of Si-crystallites could be very helpful. The in Figure 4 shown XRD-results obtain differences in the peak shapes between the conditions with nanometric Si-crystallites. A finer distribution of Si-crystallites could explain the better performance of CP T5  condition compared to the other conditions with similar Si-Phase content. 

4.)  Figure 7

    For a clearer visualization of the differences in tensile tests a modified assignment of colors and line types are suggested:

    The same heat treatment conditions should have the same color and the building directions should be separated by means of different types of lines.    

5.) Lines 225-226:

    An information about the difference in grain size (e.g. grain size number) would be very helpful for the reader. 

Author Response

(The authors gave the same response as above.)

Reviewer 3 Report

In this work, a AlSi10Mg alloy has been characterized upon selective laser melting using two strategies: using a cold as well as hot-building platform. This includes microstructure analysis (metallography, XRD), kinetic study (DSC), hardness and tensile response as a function of the building direction. Also, the aging response of as-built conditions has been investigated. The manuscript is well structured but the following aspects should be considered before publication:

I recommend avoiding acronyms i.e. SLM in the title. Also, a print strategy is ambiguous (it can include several additive manuf. Processes, for instance). Selective laser melting can include both meanings: it is clearer. 

In the abstract, at the beginning, is not clear if the thermal treatment strategies are applied by the building plate of the SLM machine or post-thermal treatment is referred. Please, specify all in the beginning for better understanding. 

Line 30.  “AlSi10Mg and AlSi7Mg alloys are amongst the most popular materials for light structures built by SLM.” This statement is ambiguous and has to be discussed in the basis of specific strength, compared to 3D printing of other metals (e.g. Ti, Ni,.. -alloys). This includes not only traditional materials but also new compositions tailored to 3D printing. Advances in both ways should be briefly mentioned in the introduction: e.g. see  Peritectic titanium alloys for 3D printing, Nat. Commun., 9, 3426 (2018),  Inducing stable alpha+beta microstructures during selective laser melting of Ti-6Al-4V using intensified intrinsic heat treatments, Materials 10,268 (2017), and Massive nanoprecipitation in an Fe-19Ni-xAl maraging steel triggered by the intrinsic heat treatment during laser metal deposition, Acta Mater. 129 (2017) 52-60.

Line 38: “during last stages of solidification”.  Is this occurring during SLM or conventional manufacturing e.g. casting? Please, specify it.

Lines 60-63: The point of this statement should be reformulated by mentioning the conclusions of the pointed studies and not the methodology employed, and how they are linked with the novelty of the present work.

Temperatures variations during cold / hot strategies have to be discussed; as well as explanation of: size of SLM  samples, region of these samples studied for the different techniques used (metallography, xrd i.e. the bulk volume measured, ....etc), size of samples for tensile testing, and if these samples were machined.

The post processing of XRD data is not described. I guess that the units for 2-Theta are degrees. They are not in the diagram. The scale used does not allow visibility of small peaks (e.g. Si peaks). Insets should be included. {} should be used indicating families of reflections (i.e. peaks)

To indicate the phases in Fig1-Fig.3 will increase readability. Also, the magnification used for comparing Fig. 1 b and Fig. 2 b is very local, and does not permit establishing conclusions (degree of refinement etc). This does not clarify that very similar microstructures lead to different properties as shown in Fig 7.

Fig.5: please specify if heating or cooling in the figure caption and exothermic direction.

Line 267: “supplying the best combination of mechanical properties”. Which ones? Tensile testing and hardness are investigated here. It is meant the strength-ductility trade-off? To specify it.

Were HP samples deformed after post-thermal treatment?  

Line 153: “fractions reported in Table 3” how are they calculated?

Line 235: “From DSC and micro-hardness results, it is inferred that HP AB samples do not show any significant response to precipitation”. Please, specify which type of precipitates is referred here and how they are linked to the resulting properties. The discussion should include how the conditions studied can lead to prone precipitation.  Characterization i.e. further evidences of the precipitates are necessary.  

The role of π-Al8Si6Mg3Fe, β-Al5SiFe, Mg2Si precipitates has to be discussed. 

Author Response

The authors would like to thank for the useful suggestions received that, in our hope, significantly contributed to improve the quality of the manuscript and to fill some gaps that were actually present in its first version.
Since several comments were raised by more than one reviewer, we prepared a single response letter were all the answers are collected.

Round  2

Reviewer 1 Report

I am happy with the modifications made by the authors.

However, there are some points need to be addressed before considering the paper for publication.

1) From line 96-100: For AlSi10Mg processed by SLM, there are more pre-heating temperatures for this alloy with respect to the values reported by the authors. 

For instance, EOS provides also 80 °C and this pre-heating is used in several works in the literature. The authors should report a more extensive description of the pre-heating temperatures used in the literature in order to give an appropriate background.  

2) Why the room temperature is reported as 35°C?

3) The authors followed the indication to indicate the reason why they selected 160 °C. 

However, the time of the pre-heating depends on the height of the job. In fact, the exposure time for the production of cubes or bars will be different. This point should be highlighted by the authors. They may report the building time for the production of tensile bars. 

4) The authors should report the standard deviation for the relative density.

In addition, the authors should provide in "materials and method" the information on the number of samples used for determining the relative density by Archimede methods. 

5) I would recommend to indicate the scanning strategy used in Materials and methods.

In fact, the scanning strategy is very important to compare the microstructure and mechanical properties with the available literature. 

Considering that they used a EOS M290, the scanning strategy should involve a rotation of 67 ° between consecutive layer.

Author Response

thank you for comments, details of our reply is given in the attache file

Reviewer 2 Report

Thank you for addressing the recommended points in your revised version and congratulations for the present paper!

Author Response

thank you for your final comment.

Maurizio Vedani